# Consistency among Office, Home, and Ambulatory Blood Pressure Values in Women with Chronic Hypertension and History of Eclampsia or Preeclampsia

**DOI:** 10.3390/jcm11175065

**Published:** 2022-08-29

**Authors:** Ewa Wojciechowska, Piotr Sobieraj, Maciej Siński, Maria Anna Zaborska-Dworak, Piotr Gryglas, Jacek Lewandowski

**Affiliations:** Department of Internal Medicine, Hypertension and Vascular Diseases, Medical University of Warsaw, 1a Banacha Street, 02097 Warsaw, Poland

**Keywords:** arterial hypertension, eclampsia, pre-eclampsia, ambulatory blood pressure monitoring

## Abstract

Adequate control of blood pressure (BP) is essential to prevent complications in pregnant women with a history of eclampsia or pre-eclampsia. However, the importance of office (OBPM), home (HBPM), and ambulatory (ABPM) BP measurements for proper control and prognosis in high-risk pregnancy is unknown. The present study aimed to compare BP values obtained during these three different BP measurements in women with a history of eclampsia or pre-eclampsia. This study included 79 pregnant women with chronic hypertension and a documented history of eclampsia or pre-eclampsia in previous pregnancy/pregnancies. Every fifth week of the study, all participants underwent ABPM, HBPM and OBPM. BP values from the 10th, 25th, and 37th weeks of pregnancy were evaluated. Therapy was intended to meet the ABPM treatment goal of <130/80 mmHg. Day, night, and 24 h ABPM systolic BP values were lower than HBPM and OBPM values at each study visit. Night and 24 h ABPM diastolic BP values were lower than HBPM and OBPM values, while day 24 h ABPM values were slightly higher than HBPM and OBPM values. ABPM provides different BP values than OBPM and HBPM. Target BP for ABPM in high-risk pregnancy hypertension should be estimated based on the predictive value of adverse pregnancy outcomes.

## 1. Introduction

Pre-eclampsia is one of the leading causes of maternal and neonatal morbidity and mortality worldwide [1,2]. The prevalence of pregnancy-related hypertension is estimated to be 10%, while the rates of eclampsia and pre-eclampsia are 0.05–1% and 2–8%, respectively [2,3]. The risk of developing pre-eclampsia is greater if the condition has occurred in previous pregnancies. In a Swedish cohort of pregnant women, the risk of developing pre-eclampsia in the first pregnancy was estimated to be 4.1%, while it was 1.7% in later pregnancies [4]. Additionally, in women with a history of pre-eclampsia in the first pregnancy, the risk of developing pre-eclampsia in the second pregnancy was 14.7%, while it was 31.9% if pre-eclampsia had occurred in previous pregnancies [4]. Arterial hypertension during pregnancy substantially increases the risk for both the fetus and the mother. Fetal risks include intra-uterine growth retardation, prematurity, and intra-uterine death, while maternal risks include placental abruption, stroke, multiple organ failure, and disseminated intravascular coagulation [5]. Early detection and appropriate management of increased blood pressure (BP) in pregnancy matter since this remains the leading cause of poor maternal, fetal, and neonatal prognosis. At present, there is a dearth of evidence that unfavorable hypertension-related outcomes in pregnancy can be decreased by lowering BP [5,6]. Irrespective of the clinical classification, the diagnosis of hypertension disorders in pregnancy is based on increased BP values. In the general population, different thresholds are used to diagnose the disease depending on the method of measuring BP [6]. Currently, hypertension in pregnancy can be diagnosed only using office BP measurements (OBPM), when BP is ≥140/90 mmHg [7]. Other methods of BP evaluation, including 24 h automatic BP monitoring (ABPM), are recommended in the management of hypertension during pregnancy. The use of ABPM is particularly useful in pregnancy due to its several advantages over other methods, including detection of white coat and masked hypertension [8,9]. ABPM is also recognized as the best method of measuring BP due its strong abilities to predict pre-eclampsia or small gestational age in pregnancy [10,11,12]. The applicability of ABPM in the management of pregnancy hypertension still remains unclear due to the lack of outcome-based studies using this method for evaluating BP [12]. This limitation is underlined in the European Society of Cardiology (ESC) guidelines regarding the therapy of pregnant women with hypertension [7]. In fact, most of the evidence regarding BP targets in pregnancy hypertension concerns OBPM values. This reduces the guidance and control of hypertension in pregnancy based on automatic or home BP measurements. Moreover, there have been no studies that selected clinical endpoints in which therapy was guided by ABPM or home BP monitoring (HBPM) in pregnant women with a history of eclampsia or pre-eclampsia. Therefore, the aim of the current study was to compare BP values measured using OBPM, HBPM, and ABPM during pregnancy in women with chronic hypertension and previous high-risk pregnancy resulting in pre-eclampsia or eclampsia.

## 2. Materials and Methods

### 2.1. Study Population

This study was conducted for a period of 4 years and included 79 pregnant women with pre-existing primary hypertension and a documented history of eclampsia or pre-eclampsia in previous pregnancy/pregnancies. Inclusion criteria were as follows: age over 18 years and the ability to attend outpatient visits every fifth week during pregnancy. Exclusion criteria were diagnosis of secondary hypertension and gestational age <4 weeks or any chronic disease demanding pharmacotherapy, including diabetes mellitus, autoimmune disease, or chronic kidney disease. This study was approved by the local ethics committee (No. AKBE/71/2018).

Every 5th week of the study, all participants underwent ABPM and OBPM with the last visit scheduled in the 37th week of pregnancy. Before each visit, HBPM measurements were performed. For this study, BP measurements from 10th, 25th, and 37th weeks of pregnancy were evaluated.

### 2.2. Office Blood Pressure Measurement

OBPM was performed during each visit with an automatic oscillometric device (Omron HEM-7120-E, Omron Healthcare Co., Ltd, Kyoto, Japan) after 5 min of rest, as recommended by the ESC guidelines [6]. The patient was placed in a sitting position with the legs uncrossed and arm supported at the level of the heart. Cuff size was fitted accordingly to the circumference of the patient’s arm. Participants were asked to abstain from drinking coffee and smoking before BP measurements.

### 2.3. Ambulatory Blood Pressure Measurement

The day before each medical visit, 24 h ABPM was performed with a SpaceLabs 90217A device according to the ESC guidelines [6]. Measurements were taken every 15 min during the daytime (from 6:00 a.m. to 10:00 p.m.) and every 30 min during the night-time (from 10:00 p.m. to 6:00 a.m.). All participants were instructed to follow their normal daily routine. Recordings with at least 25 measurements during the day and 10 at night were considered for further evaluation.

### 2.4. Home Blood Pressure Measurement

Within 7 days before each medical visit, HBPM was performed with an oscillometric device. Only certified electronic (Omron Hem-9210T) devices were used. All study participants were trained to follow the principles of home BP measurement. Type and cuff size of the devices and reliability of the measurements were verified. HBPM was performed twice in the morning and twice in the evening for 3–7 days before each study visit. The results of HBPM were computed as mean of systolic and diastolic BP values obtained during the measurements.

### 2.5. Treatment

In all subjects, methyldopa, labetalol, or nifedipine were used. Some females also used metoprolol transiently. The antihypertensive drugs were administered according to the following principles: first methyldopa, then labetalol, followed by nifedipine, as needed. The doses of antihypertensive drugs were titrated to reach and maintain mean 24 h SBP/DBP below 130/80 mmHg. In cases of significant discrepancy in BP values between ABPM and OBPM, doses were changed based on the former method. The dose of antihypertensive drug was also titrated up when the maximal OBPM or HBPM value was higher than 160/105 mmHg.

At the end of the study, 25% of the females had received one antihypertensive drug, 55% had received two drugs, and 20% had received three drugs. All subjects also received 75 mg of acetylsalicylic acid once daily.

### 2.6. Study Outcomes

The primary study outcome was the agreement among ABPM, OBPM, and HBPM in women with a high-risk pregnancy. The secondary clinical endpoint was an adverse pregnancy outcome defined as follows: low birth weight (defined as <2500 g), preterm birth (defined as <37 weeks), and Apgar score.

### 2.7. Statistical Analysis

Quantitative variables are presented as mean followed by standard deviation or as median with minimum and maximum range depending on the data distribution. The data were tested to confirm that they followed a normal distribution using Shapiro–Wilk test. Discrete variables are presented as number and percentage.

The agreement between BP measurements obtained during ABPM, HBPM, and OBPM methods was assessed using Bland–Altman plots and statistics. This method of analysis consists in calculating the differences and means of the values obtained by the two measurement methods for each subject. Then differences and means are presented using scatterplots. Bias between methods is calculated as the mean of the differences. The upper and lower agreement limits are also calculated assuming 95% of the calculated differences will be within their range. In our study the bias, upper and lower limits of agreement (LoA) were computed and presented with 95% confidence intervals. Statistical analysis was performed using the environment for statistical computing R 3.6.0 (R Foundation for Statistical Computing, Vienna, Austria) with the ’blandr’ package.

## 3. Results

### 3.1. Characteristics of the Study Group

The study included 79 pregnant women aged 34 ± 4.7 years. During previous pregnancies, pre-eclampsia had occurred in 71 (89.9%) patients and eclampsia in 8 (10.1%). Ten (12.7%) women had a history of stillbirth and 72 (91.1%) had a history of miscarriage during previous pregnancies. Pre-existing hypertension was found in 62 women (78.5%). In 17 women (21.5%), hypertension was diagnosed during prior pregnancy, but at the beginning of the study they were normotensive. Mean pre-pregnancy weight was 66.1 ± 10.2 kg, while mean pre-pregnancy BMI was 24.1 ± 3.2 kg/m^2^.

The baseline characteristics of the analyzed population and pregnancies are presented in Table 1. In the analyzed group, 59 (74.7%) women were in their second pregnancy, 11 (13.9%) were in their third pregnancy, and 9 (5.1%) were in their fourth or later pregnancy. Overall, 78 (98.7%) women had singleton pregnancies and 1 (1.3%) had a twin pregnancy. During the study, 36 female and 44 male offspring were born.

### 3.2. Achieved Blood Pressure

BP values measured in the 10th, 25th, and 37th weeks of pregnancy are presented in Table 2. The blood pressure was well controlled during the study. Day, resting, and 24 h ABPM SBP values were lower than HBPM and OBPM values at each study visit. Resting and 24 h ABPM DBP values were lower than HBPM and OBPM values, while day 24 h values were slightly higher than the BP during HBPM and OBPM. The agreement between systolic and diastolic BP measurements was calculated for all methods of BP measurements taken during each visit, as presented in Table 3.

### 3.3. Systolic Blood Pressure

The worst agreement of measured SBP values (bias ~10 mmHg) was found upon comparing 24 h ABPM and OBPM. SBP values obtained as HBPM were closer to 24 h ABPM values (bias ~5.5 mmHg). The best agreement of BP measurements was between day ABPM and HBPM values (bias ~2.5 mmHg). The level of agreement between the methods was similar at each study visit (Figure 1, Figure 2 and Figure 3, Table 3).

### 3.4. Diastolic Blood Pressure

Better agreement between methods was found when DBP instead of SBP measurement was considered. Similarly, the worst agreement was seen when 24 h ABPM values were compared with OBPM (~5 mmHg). Day ABPM delivered higher values than HBPM at each study visit (Figure 1, Figure 2 and Figure 3, Table 3).

### 3.5. Pregnancy Outcomes

The surveyed women gave birth on average at 38 ± 2.1 weeks of gestation. Preterm delivery was present in 12 (15.2%) subjects. There were no stillbirths or miscarriages. Median newborn birth weight was 3300 g (interquartile range 3100–3565 g). Only five (6.3%) subjects had low birth weight (<2500 g); however, one newborn had extremely low birth weight (<1000 g) of 955 g. Mean Apgar score was 9.9 (2 newborns had 8 points, 6 newborns had 9 points, and the others had 10 points).

## 4. Discussion

The current study estimates the agreement among ABPM, OBPM, and HBPM in women with a history of chronic hypertension and a history of eclampsia or pre-eclampsia in previous pregnancies. All three methods of BP measurement are currently recommended as being suitable for the diagnosis and management of hypertension [6,7]. Currently, OBPM is the most commonly used method, which allows for cardiovascular risk assessment and treatment initiation [6]. Owing to the fact that OBPM must be performed in a clinical setting, the application of this method in long-term follow-up is limited. Out-of-office BP measurement with ABPM and HBPM may be an option to confirm the diagnosis of hypertension and the accurate assessment of untreated and treated patients [13]. For economic reasons, ABPM is not commonly available, while HBPM may be useful for the long-term evaluation of patients. The roles of both ABPM and HBPM in detecting white coat and masked hypertension are often emphasized [6,13]. Interestingly, the roles of ABPM and HBPM have not been fully assessed in some groups of patients, including obese subjects, adolescents, or pregnant women. In addition, in some populations, appropriate BP thresholds and targets for HBPM and ABPM remain unknown [6,7]. In particular, there is a lack of outcome-based comparisons between office-BP- and out-of-office-BP-guided treatment [6].

In the current study, it was shown that BP values in ABPM were lower than BP values in OBPM. Except for the agreement between day ABPM DBP and HBPM DBP values, ABPM also provided lower BP values than HBPM. The best agreement between methods was observed when day ABPM SBP and DBP values were compared with the corresponding HBPM values. However, all comparisons for which agreement was identified were associated with a large range between the upper and lower LoA.

The major strength of this study is the fact that the compatibility between different methods of BP measurement was assessed in a specific population, namely, in women with a history of pre-eclampsia or eclampsia in previous pregnancies. Owing to the unique characteristics of the study group and the lack of previous studies in similar groups of patients, it is difficult to compare the current results with the results of other researchers. To the best of our knowledge, there is a lack of studies evaluating BP values measured by these three methods in similar populations of patients. In a study by Brown et al., ABPM and OBPM values were compared in normal pregnancy [14]. Contrary to our results, they found that the ABPM SBP values were lower by 10 ± 5 mmHg and the ABPM DBP values by 6 ± 6 mmHg than corresponding OBPM indices. In another study by Churchill et al., BP in ABPM and OBPM in the 18th, 28th, and 36th weeks of pregnancy were compared in a small group of pregnant women without hypertension at the beginning of the trial. In this study, BP values during 24 h ABPM were also significantly higher by 5 and 5.5 mmHg at 18 weeks, by 3 and 6.5 mmHg at 28 weeks, and by 5 and 5.5 mmHg at 36 weeks than BP obtained during OBPM [15]. ABPM was also compared with self-initiated BP measurement (using Omron HEM 705 CP device) and conventional measurement using a mercury sphygmomanometer [16]. When day ABPM SBP and DBP values and self-initiated BP measurement device values were compared, no differences were found by paired *t* test. Day ABPM DBP values were however lower than DBP measured using a mercury sphygmomanometer (77 ± 7 versus 80 ± 7 mmHg, *p* < 0.001). When Bland–Altman analysis was used to assess the agreement between day ABPM and self-initiated device BP values, relatively small bias (for SBP 0.2 mmHg and for DBP −0.1 mmHg) with a wide range between upper and lower LoA was reported (for SBP 32.4 mmHg and for DBP 24.9 mmHg). Additionally, higher bias (for SBP −1.2 mmHg and DBP −3.4 mmHg) was reported when the agreement between day ABPM and mercury sphygmomanometer values was calculated and a wide range between the LoA was reported. Values of BP obtained using ABPM were also compared with single and multiple (5 times per 24 h) BP measurements using a mercury sphygmomanometer in 99 pregnant women hospitalized for hypertension [17]. It was found that ABPM single and multiple BP recordings showed different values (ABPM SBP higher by 3.1 mmHg than multiple measurements and by 4.6 mmHg than single BP measurement, with ABPM DBP lower by 2.4 mmHg and by 2.7 mmHg, respectively). Moreover, the authors underlined that, despite relatively small differences between BP values obtained with these two methods, ABPM and traditional BP measurement cannot be used interchangeably.

HBPM is a common method of BP measurement in the general population and the BP values obtained by this method are close to ABPM; to the best of our knowledge, no direct comparisons between the two methods in a population of pregnant women have been performed. Owing to the increasing use of HBPM, its detailed comparison with ABPM as another out-of-office method has been presented elsewhere [18]. Several articles regarding agreement between the two methods are worth mentioning, although their results are inconclusive. Chrubasik et al. pointed out that HBPM delivered lower SBP (by 3.65 ± 10.3 mmHg) and higher DBP (by 3.15 ± 6.4 mmHg) values when compared to 24 h ABPM at the beginning of the study [19]. HBPM values were higher than both daytime SBP and DBP in ABPM by 3.8 ± 3.7 and 2.5 ± 1.8 mmHg, respectively. However, in this study, Bland–Altman analyses for comparison of the agreement were performed. Similar results were presented in a recent study where HBPM and ABPM were compared in children, adolescents, and young adults [20]. Both SBP and DBP values in HBPM were lower than daytime ABPM BP values by 4.8 ± 7.4 mm Hg and 3.4 ± 5.6 mm Hg, respectively. In addition, Eguchi et al. compared HBPM and ABPM in a group of 56 adults using Bland–Altman analysis [21]. They showed that the mean difference between day ABPM SBP and HBPM SBP was 2.2 ± 10.8 mmHg. The agreement between DBP measurements was similar—the mean difference between HBPM DBP and day ABPM DBP was 2.2 ± 7.8 mmHg.

Finally, Ntineri et al. reported important data evaluating the agreement between HBPM and ABPM [22]. After BP assessment by ABPM and HBPM in 1971 participants, it was shown that SBP/DBP in HBPM was significantly higher (by 3.3 ± 10.5/3.8 ± 6 mmHg) than the corresponding values in 24 h ABPM but lower (by 1.4 ± 10.8/0.01 ± 6.4 mmHg) than day ABPM. Contrary to Oloffson, the authors concluded that there was considerable diagnostic agreement between HBPM and ABPM and that these methods were interchangeable for clinical decisions in most patients [17,22].

When describing the results of comparisons between different methods of BP measurement performed by some researchers, it should be noted that the agreement between methods should be assessed by Bland–Altman analysis [23,24]. In a significant part of the research, such an analysis was not used, which prompts a discussion about statistical methods and requires reflection on the conclusions drawn by the authors [25].

Our study has several limitations. We observed that most childbirths took place on the estimated date of delivery and there were no stillbirths or miscarriages. Both median newborn birth weight and mean Apgar score were within normal limits. However, we cannot state that the favorable outcome of pregnancy in the investigated group was related to satisfactory ABPM-guided BP control due to the lack of a control group treated on the basis of OBPM. However, the main aim of this study was to report the agreement between most available BP measurements instead of evaluating the impact of BP control on pregnancy outcome. In the current study, neither attended nor unattended office BP measurements were investigated. Some data showed that observer presence could substantially influence the study results [26]. Additionally, it was found that some factors may affect HBPM/ABPM differences, including age, BMI, history of cardiovascular disease, office hypertension, and antihypertensive treatment [21]. In particular, antihypertensive drug treatment may have independently increased the odds of diagnostic disagreement in our study.

## 5. Conclusions

In conclusion, ABPM provides different BP values than OBPM and HBPM. These differences have been identified earlier and have now been confirmed in females with chronic hypertension and history of high-risk pregnancy. If only the bias between methods is considered, the differences are small, but according to the LOA range, the differences may be essential in a particular patient. Therefore, our results are not sufficient to determine normal BP range for ABPM and this range should be established on the basis of the predictive value of an unfavorable pregnancy result.

## Figures and Tables

**Figure 1 jcm-11-05065-f001:**
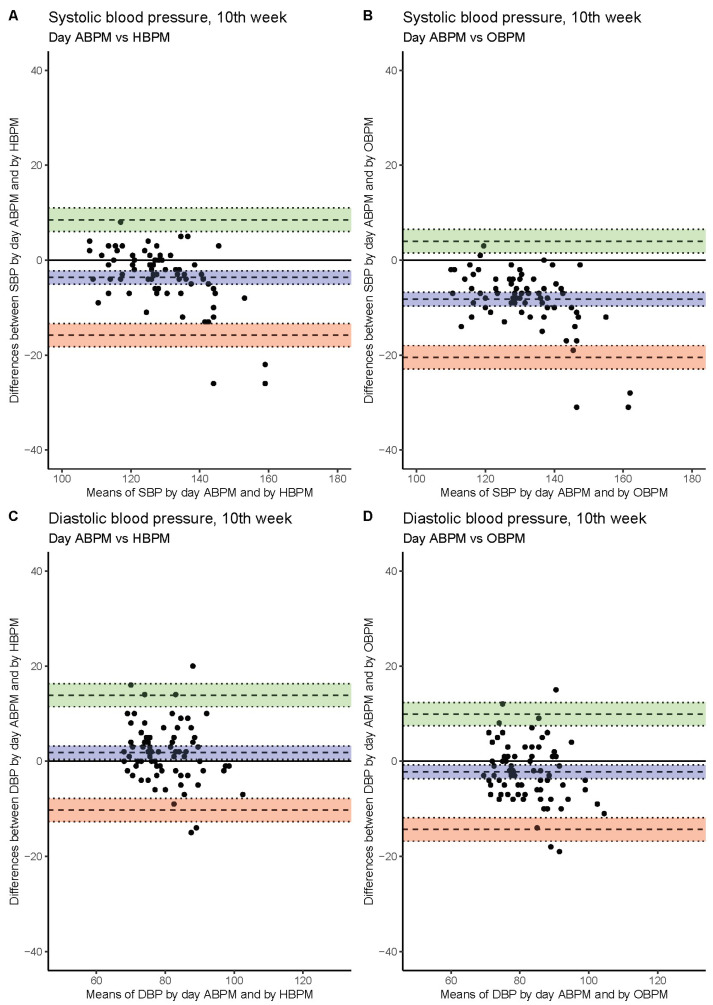
The agreement between systolic blood pressure values obtained by day ABPM vs. HBPM (**A**) and day ABPM vs. OBPM (**B**). Diastolic blood pressure measured by day ABPM vs. HBPM (**C**) and day ABPM vs. OBPM (**D**) in 10th week of gestation. For each patient difference and mean of blood pressure values obtained by two methods was calculated and plotted (black dots). Mean of differences is presented as bias with computed 95% confidence interval (blue). Upper and lower limit of agreement with 95% confidence intervals are presented using green and pink colors, respectively.

**Figure 2 jcm-11-05065-f002:**
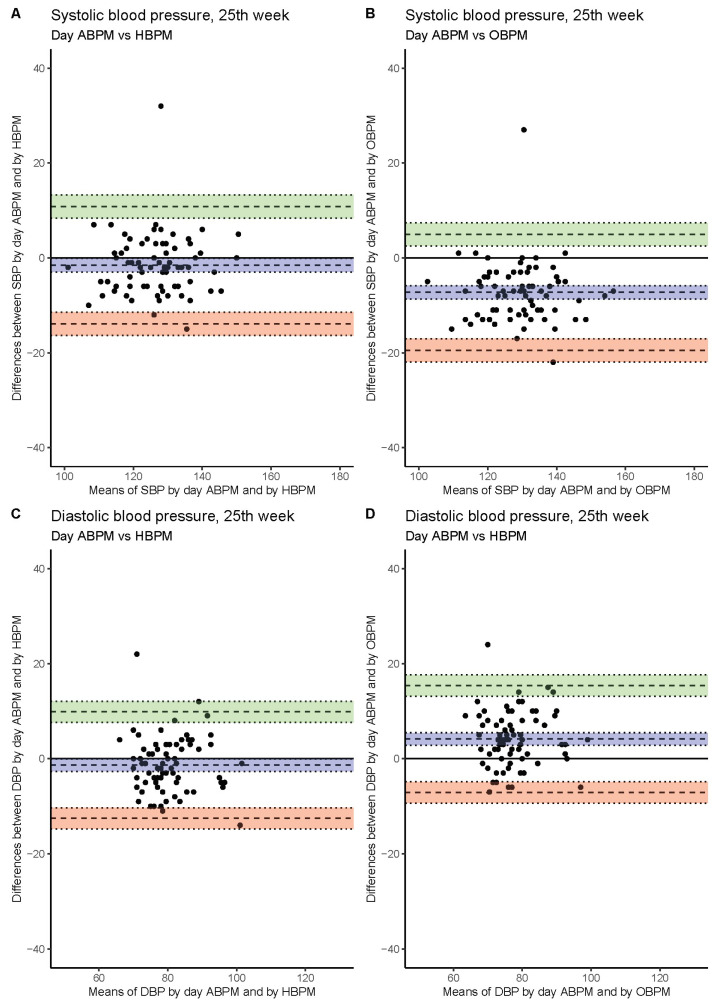
The agreement between systolic blood pressure values obtained by day ABPM vs. HBPM (**A**) and day ABPM vs. OBPM (**B**). Diastolic blood pressure measured by day ABPM vs. HBPM (**C**) and day ABPM vs. OBPM (**D**) in 25th week of gestation. For each patient difference and mean of blood pressure values obtained by two methods was calculated and plotted (black dots). Mean of differences is presented as bias with computed 95% confidence interval (blue). Upper and lower limit of agreement with 95% confidence intervals are presented using green and pink colors, respectively.

**Figure 3 jcm-11-05065-f003:**
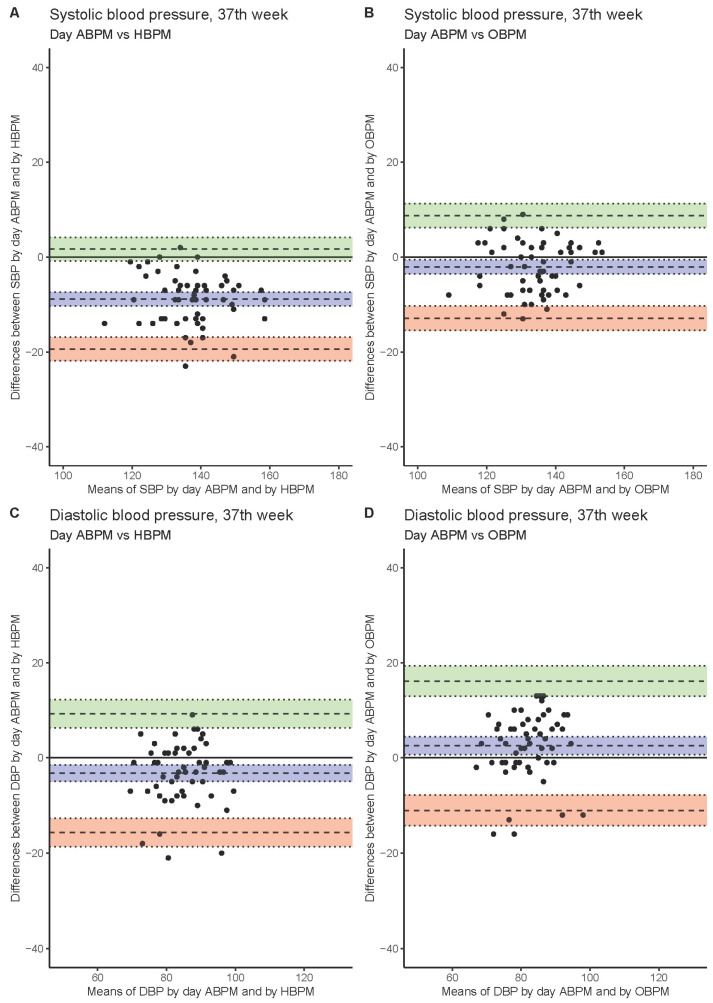
The agreement between systolic blood pressure values obtained by day ABPM vs. HBPM (**A**) and day ABPM vs. OBPM (**B**). Diastolic blood pressure measured by day ABPM vs. HBPM (**C**) and day ABPM vs. OBPM (**D**) in 37th week of gestation. For each patient difference and mean of blood pressure values obtained by two methods was calculated and plotted (black dots). Mean of differences is presented as bias with computed 95% confidence interval (blue). Upper and lower limits of agreement with 95% confidence intervals are presented using green and pink colors, respectively.

**Table 1 jcm-11-05065-t001:** Clinical characteristic and pregnancy outcomes.

Parameter	Value (*N* = 79)
Age (years)	34 ± 4.7
Baseline blood pressure (mm Hg):	
24 h ABPM SBP/DBP	122.9 ± 13.1/77.0 ± 9.9
Day ABPM SBP/DBP	126.9 ± 13.6/80.9 ± 10.4
Night ABPM SBP/DBP	113.7 ± 13.8/67.9 ± 9.9
OBPM SBP/DBP	134.0 ± 15.9/83.7 ± 11.4
HBPM SBP/DBP	128.1 ± 16.5/79.9 ± 11.3
Pre-pregnancy nutritional status:	
BMI (kg/m^2^)	24.1 ± 3.2
Underweight (*n*, %)	1 (1.3)
Normal weight (*n*, %)	50 (63.3)
Overweight (*n*, %)	28 (35.4)
Obese (*n*, %)	1 (1.3)
History of pre-eclampsia (*n*, %)	71 (89.9)
History of eclampsia (*n*, %)	8 (10.1)
History of stillbirth (*n*, %)	10 (12.7)
History of miscarriage (*n*, %)	72 (91.1)
Obstetric history:	
2nd pregnancy (*n*, %)	59 (74.7)
3rd pregnancy (*n*, %)	11 (13.9)
4th pregnancy (*n*, %)	4 (5.1)
5th pregnancy (*n*, %)	3 (3.8)
6th pregnancy (*n*, %)	1 (1.3)
8th pregnancy (*n*, %)	1 (1.3)

ABPM—ambulatory blood pressure monitoring, HBPM—home blood pressure monitoring, OBPM—office blood pressure measurement, SBP—systolic blood pressure, DBP—diastolic blood pressure, BMI—body mass index.

**Table 2 jcm-11-05065-t002:** Blood pressure values during pregnancy according to the period of the measurement (10th, 25th, 37th week) and method of blood pressure estimation. All comparisons of blood pressure values were made with ABPM 24 h systolic or diastolic blood pressure, respectively.

Pregnancy Week	Method of Measurement	SBP (mm Hg)	*p*-Value	DBP (mm Hg)	*p*-Value
10th	ABPM 24 h	123.5 ± 10.7	-	77.6 ± 8	-
	ABPM day	126.9 ± 10.1	<0.001	81.2 ± 7.8	<0.001
	ABPM night	114.7 ± 12.3	<0.001	69.6 ± 9.3	<0.001
	HBPM	130.6 ± 13.5	<0.001	79.4 ± 9.1	0.057
	OBPM	135.2 ± 13.6	<0.001	83.4 ± 9.3	<0.001
25th	ABPM 24 h	122.4 ± 10	-	76.3 ± 7.6	-
	ABPM day	125.8 ± 10.3	<0.001	79.6 ± 7.9	<0.001
	ABPM night	114.4 ± 11.5	<0.001	68.3 ± 8.8	<0.001
	HBPM	127.3 ± 10.1	<0.001	75.5 ± 7.8	0.728
	OBPM	133.1 ± 10.5	<0.001	81 ± 8.1	<0.001
37th	ABPM 24 h	129.3 ± 9.4	-	79.9 ± 7.6	-
	ABPM day	132.7 ± 9.9	<0.001	83.3 ± 8.2	<0.001
	ABPM night	122.6 ± 13.8	<0.001	72.1 ± 7.9	<0.001
	HBPM	134.8 ± 9.6	<0.001	80.7 ± 7.3	1.0
	OBPM	141.5 ± 10.5	<0.001	86.5 ± 7.9	<0.001

HBPM, OBPM, ABPM—home, office, ambulatory blood pressure measurements. All *p*-values are adjusted for multiple comparisons according to Bonferroni formula.

**Table 3 jcm-11-05065-t003:** The agreement between ABPM, HBPM, and OBPM measurements of blood pressure in 10th, 25th, and 37th week of pregnancy assessed using Bland–Altman method.

Week	Method of Blood Pressure Measurement	Bias (mm Hg)	Lower Limit of Agreement (mm Hg)	Upper Limit of Agreement (mm Hg)
10th	SBP: 24 h ABPM vs. HBPM	−7 (−8.6–−5.5)	−20.4 (−23.1–−17.7)	6.3 (3.6–9)
	SBP: 24 h ABPM vs. OBPM	−11.6 (−13.2–−10.1)	−24.9 (−27.6–−22.3)	1.7 (−1–4.4)
	DBP: 24 h ABPM vs. HBPM	−1.8 (−3.2–−0.4)	−14 (−16.4–−11.5)	10.4 (7.9–12.8)
	DBP: 24 h ABPM vs. OBPM	−5.8 (−7.2–−4.4)	−17.9 (−20.4–−15.5)	6.3 (3.8–8.7)
	SBP: day ABPM vs. HBPM	−3.7 (−5.1–−2.2)	−15.8 (−18.3–−13.4)	8.5 (6.1–11)
	SBP: day ABPM vs. OBPM	−8.2 (−9.7–−6.8)	−20.5 (−22.9–−18)	4 (1.5–6.5)
	DBP: day ABPM vs. HBPM	1.8 (0.4–3.2)	−10.3 (−12.7–−7.8)	13.9 (11.4–16.3)
	DBP: day ABPM vs. OBPM	−2.2 (−3.7–−0.8)	−14.4 (−16.8–−11.9)	9.9 (7.5–12.3)
	SBP: OBPM vs. HBPM	4.6 (4.3–4.9)	2.2 (1.7–2.6)	7 (6.5–7.5)
	DBP: OBPM vs. HBPM	4 (3.6–4.4)	0.7 (0–1.4)	7.3 (6.7–8)
25th	SBP: 24 h ABPM vs. HBPM	−4.9 (−6.1–−3.8)	−14.6 (−16.5–−12.7)	4.7 (2.8–6.6)
	SBP: 24 h ABPM vs. OBPM	−10.7 (−11.7–−9.6)	−19.9 (−21.7–−18)	−1.5 (−3.3–0.4)
	DBP: 24 h ABPM vs. HBPM	0.8 (−0.4–2.1)	−9.8 (−11.9–−7.7)	11.4 (9.3–13.5)
	DBP: 24 h ABPM vs. OBPM	−4.7 (−5.9–−3.5)	−15 (−17.1–−13)	5.7 (3.6–7.8)
	SBP: day ABPM vs. HBPM	−1.5 (−3–−0.1)	−13.9 (−16.4–−11.4)	10.8 (8.4–13.3)
	SBP: day ABPM vs. OBPM	−7.3 (−8.7–−5.8)	−19.5 (−21.9–−17)	5 (2.5–7.4)
	DBP: day ABPM vs. HBPM	4.1 (2.8–5.4)	−7.1 (−9.3–−4.9)	15.4 (13.1–17.6)
	DBP: day ABPM vs. OBPM	−1.4 (−2.6–−0.1)	−12.5 (−14.8–−10.3)	9.8 (7.6–12.1)
	SBP: OBPM vs. HBPM	5.7 (5.4–6.1)	2.9 (2.3–3.5)	8.5 (8–9.1)
	DBP: OBPM vs. HBPM	5.5 (5–6)	1.5 (0.7–2.3)	9.5 (8.7–10.3)
37th	SBP: 24 h ABPM vs. HBPM	−5.5 (−6.9–−4)	−15.9 (−18.4–−13.5)	5 (2.5–7.4)
	SBP: 24 h ABPM vs. OBPM	−12.3 (−13.8–−10.8)	−23.1 (−25.6–−20.5)	−1.5 (−4.1–1.1)
	DBP: 24 h ABPM vs. HBPM	−0.9 (−2.7–1)	−14.2 (−17.3–−11)	12.4 (9.3–15.6)
	DBP: 24 h ABPM vs. OBPM	−6.6 (−8.3–−4.9)	−18.9 (−21.8–−16)	5.7 (2.7–8.6)
	SBP: day ABPM vs. HBPM	−2.1 (−3.6–−0.6)	−12.9 (−15.5–−10.3)	8.7 (6.2–11.3)
	SBP: day ABPM vs. OBPM	−8.9 (−10.3–−7.4)	−19.4 (−21.9–−16.9)	1.7 (−0.8–4.2)
	DBP: day ABPM vs. HBPM	2.5 (0.7–4.4)	−11 (−14.3–−7.8)	16.1 (12.9–19.3)
	DBP: day ABPM vs. OBPM	−3.2 (−4.9–−1.5)	−15.7 (−18.6–−12.7)	9.2 (6.3–12.2)
	SBP: OBPM vs. HBPM	6.8 (6–7.6)	0.8 (−0.6–2.2)	12.8 (11.3–14.2)
	DBP: OBPM vs. HBPM	5.8 (5–6.5)	0.5 (−0.7–1.8)	11 (9.8–12.3)

ABPM—ambulatory blood pressure monitoring, HBPM—home blood pressure monitoring, OBPM—office blood pressure measurement, SBP—systolic blood pressure, DBP—diastolic blood pressure.

## Data Availability

Due to further pending analyses, the data are not currently available.

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
