# Peer review of "Consistency among Office, Home, and Ambulatory Blood Pressure Values in Women with Chronic Hypertension and History of Eclampsia or Preeclampsia"

_jcm, 2022, doi:10.3390/jcm11175065_

Round 1
Reviewer 1 Report
The paper describes a prospective study comparing BP measurements obtained in-office (OBPM), at-home (HBPM), and by 24h ambulatory BP device (ABPM) in pregnant patients with chronic HTN and a history of preeclampsia. The principal findings are that some devices tend to read higher or lower than other devices at the measured time-points during pregnancy.
The findings are novel in that such comparison has not been reported in this specific population of pregnant patients, but such comparisons have been extensively made in the general medicine literature for non-pregnant patients. The writing is sufficiently clear. The use of Bland-Altman plots for the primary statistical method is appropriate for this type of study.
Some points needing attention:
1. The description of the study population in the title (“high-risk pregnancy”) is vague. The description in the abstract and on line 63 (“women with a history of preeclampsia or eclampsia”) is inadequate. The population was women who had BOTH chronic hypertension and a history of preeclampsia or eclampsia. The description on lines 194-195 and 297 (“hypertensive women”) is not accurate because many of them were normotensive during the study. Better to call them “women with chronic hypertension”.
2. Throughout the paper, “awake” is not a proper description of the daytime ABPM measurements because the methods do not specify that there was any attempt to determine the sleep-wake status of the patients at the time of measurement. Better to call them “day” measurements.
3. Line 61 implies that the principal aim of the study was to establish reliable and safe threshold of ABPM, but line 115 states this was a secondary aim. The clinical endpoints used are not well-chosen for this population. In women with CHTN and prior preeclampsia, a mandatory endpoint would be development of recurrent preeclampsia. Preterm birth cannot be understood in isolation without knowing how many were spontaneous, how many were indicated, and in this population, how many were indicated because of hypertensive disorders. Low birthweight cannot be interpreted by itself since it can be attributed to preterm birth in a normally grown fetus or to fetal growth restriction. Apgar score is not an outcome, it is an indicator of the need for resuscitation.
4. There appears to have been no attempt to correlate BP measurements with the “adverse outcomes” listed. It is unclear to me how anything about the study design would yield any information about “reliable and safe thresholds” of ABPM.
5. Many clinical readers will be unfamiliar with Bland-Alman plots and their interpretation. As presented, interpretation of the figures requires a huge amount of work by the reader. Revision of the figures or the captions is strongly recommended:
a. There needs to be some sort of legend or key explaining that the green band represents the upper LOA and CI, the purple band the “bias” and CI, and the pink band the lower LOA and CI. This can appear in every panel or on one panel per figure. Alternatively, it can be explained in the caption.
b. The x-axis title must show which mean is being plotted, e.g. “Mean Home Blood Pressure, mmHg” and the units of measurement
c. The y-axis title must show which difference is being plotted e.g. “Day ABPM minus HBPM, mmHg” and the units of measurement
d. The caption must explain that each dot represents the mean values from 1 patient (n=79). If that’s not what each dot represents, then please explain what is shown.
e. Labeling of the panels as A, B, C, D requires the reader to scan repeatedly up and down between the figure and the caption to understand what is being shown. I got dizzy going up and down repeatedly to figure this out. Far preferable if you would label the panels descriptively; for example, Panel A in Figure 1 could be labeled “Systolic, Day ABPM vs HBPM, 10 weeks”, Panel B “Systolic, Day ABPM vs OBPM, 10 weeks” etc.
Author Response
Reviewer 1.
Thank you very much to the reviewer for the comments submitted. We agree with all comments. Below, after the comments of the reviewer, we present our answers. Accordingly, appropriate changes were made to the manuscript. Changes to the text of manuscript were marked up using the
“Track Changes” function.
Some points needing attention:
1.The description of the study population in the title (“high-risk pregnancy”) is vague. The description in the abstract and on line 63 (“women with a history of preeclampsia or eclampsia”) is inadequate. The population was women who had BOTH chronic hypertension and a history of preeclampsia or eclampsia. The description on lines 194-195 and 297 (“hypertensive women”) is not accurate because many of them were normotensive during the study. Better to call them “women with chronic hypertension”.
According to the Reviewer comment we changed title of the manuscript: “Consistency among office, home, and ambulatory blood pressure values in women with chronic hypertension and history of eclampsia or preeclampsia”. Additionally in the abstract in description of investigated population we added word: “chronic (hypertension)”. Also in Introduction we added: “chronic hypertension and” describing our population. In discussion we added: “in women with a history of chronic hypertension” and in conclusion “with chronic hypertension and history of high-risk pregnancy”.
2. Throughout the paper, “awake” is not a proper description of the daytime ABPM measurements because the methods do not specify that there was any attempt to determine the sleep-wake status of the patients at the time of measurement. Better to call them “day” measurements.
According to the Reviewer suggestion we replaced word “awake” with “day”.
3. Line 61 implies that the principal aim of the study was to establish reliable and safe threshold of ABPM, but line 115 states this was a secondary aim. The clinical endpoints used are not well-chosen for this population. In women with CHTN and prior preeclampsia, a mandatory endpoint would be development of recurrent preeclampsia. Preterm birth cannot be understood in isolation without knowing how many were spontaneous, how many were indicated, and in this population, how many were indicated because of hypertensive disorders. Low birthweight cannot be interpreted by itself since it can be attributed to preterm birth in a normally grown fetus or to fetal growth restriction. Apgar score is not an outcome, it is an indicator of the need for resuscitation.
The principal aim of the study presented in introduction was corrected and therefore a part of sentence: “established reliable and safe BP thresholds in ABPM” was removed. We absolutely agree with the Reviewer on the choice and importance of secondary endpoints. In the manuscript, we should not described them as secondary endpoints, but rather as parameters which characterize the course of pregnancy and its effect. Hence, we have removed the sentence referring to secondary endpoints from the Study outcomes section.
4. There appears to have been no attempt to correlate BP measurements with the “adverse outcomes” listed. It is unclear to me how anything about the study design would yield any information about “reliable and safe thresholds of ABPM.
As rightly noted by the reviewer and which was removed from the manuscript, the purpose of the study was not to assess reliable and safe thresholds "in ABPM
5. Many clinical readers will be unfamiliar with Bland-Alman plots and their interpretation. As presented, interpretation of the figures requires a huge amount of work by the reader. Revision of the figures or the captions is strongly recommended:
We agree with the Reviewer and changed the plots and labels as below. We also added in ‘Statistical analysis’ subsection short explanation of Bland-Altman analysis for non-familiar with this method readers: “This method of analysis consists in calculating the differences and means of the values obtained by the two measurement methods for each subject. Then differences and means are presented using scatterplot. Bias between methods is calculated as the mean of the differences. The upper and lower agreement limits are also calculated assuming 95% of the calculated differences will be within their range.”
a. There needs to be some sort of legend or key explaining that the green band represents the upper LOA and CI, the purple band the “bias” and CI, and the pink band the lower LOA and CI. This can appear in every panel or on one panel per figure. Alternatively, it can be explained in the caption.
We added an explanation in each caption of the figures as follows: ”For each patient difference and mean of blood pressure values obtained by two methods was calculated and plotted (black dots). Mean of differences is presented as bias with computed 95% confidence interval (blue). Upper and lower limit of agreement with 95% confidence intervals are presented using green and pink color, respectively.”
b. The x-axis title must show which mean is being plotted, e.g. “Mean Home Blood Pressure, mmHg” and the units of measurement
We added proper label to x-axis of each plot.
c. The y-axis title must show which difference is being plotted e.g. “Day ABPM minus HBPM, mmHg” and the units of measurement
We added proper label to y-axis of each plot.
d. The caption must explain that each dot represents the mean values from 1 patient (n=79). If that’s not what each dot represents, then please explain what is shown.
We added explanation in captions.
e. Labeling of the panels as A, B, C, D requires the reader to scan repeatedly up and down between the figure and the caption to understand what is being shown. I got dizzy going up and down repeatedly to figure this out. Far preferable if you would label the panels descriptively; for example, Panel A in Figure 1 could be labeled “Systolic, Day ABPM vs HBPM, 10 weeks”, Panel B “Systolic, Day ABPM vs OBPM, 10 weeks” etc.
We added explanations to each plot according to Reviewer suggestion.
Reviewer 2 Report
Dear author,
This is a good research topic and I see that quite a good work has been done to bring out the results of the study which may benefit readers in having some evidence for ambulatory BP measurements. I would like to draw your attention to some minor and major changes for the present format which you may accept if you feel necessary and appropriate. CLIP trials could give an input on the benefits of repeated BP measurements at community level and also the device used are more validated in PIH.
1. Introduction:
1a. Page 2, Line 61. It should be establish instead of established.
1b. Are the abbreviations used standard abbreviations?
2. Materials and Methods:
2a. Duration of the study not mentioned
2b. Line 69, Inclusion criteria were as follows: age over 18 years to be changed to Inclusion criteria were age over 18 years...
-2c. Inclusion/ Exclusion Criteria: Looks incomplete, since the results does not include any nulliparous or women with first pregnancy. It looks like there is some study selection criteria which excludes these women. From Table 1- obstetric history, it is evident that women with 2nd, 3rd, 4th, 5th, 6th or 8th pregnancy were included in the study. In the introduction there is a mention of a Swedish study stating that risk of preeclampsia is more in first pregnancy. Kindly explain why women with first pregnancy were not included.
-2d. Even though BP measurements were taken every 5th week, why only three readings (10th, 25th and 37th week) were considered for analysis? Rationale behind this?
-2e. Are these devices from Omeron and Spacelabs been validated and approved for measuring pregnancy induced hypertension?
-2f. For HBPM; what measures were taken to ensure that the BP measurement procedures were followed correctly? The two measurements in the morning and evening- were they done with some time interval between them? If so, how much?
-2g. Treatment: Was this a part of the study procedure? If so, what is its relevance in this study keeping in account the study objectives?
3. Results
3a. Pre-pregnancy data on weight and BMI. Was this self reported data by the women or was it measured? If measured, at what point were the women approached and included in the study?
-3b. No mention of why women with first pregnancy not included in the study.
-3c. Figures: All these figures may not be required as the data is already presented in the Table 3. Please recheck if you can put only the relevant figures.
4. Discussion: Suggestion to improve the presentation in discussion section so as to appeal the readers and focusing more on the objectives of the study.
Author Response
Reviewer 2.
Thank you very much to the reviewer for the comments submitted. We agree with all comments. Below, after the comments of the reviewer, we present our answers. Accordingly, appropriate changes were made to the manuscript. Changes to the text of manuscript were marked up using the
“Track Changes” function.
Comments and Suggestions for Authors
Dear author,
This is a good research topic and I see that quite a good work has been done to bring out the results of the study which may benefit readers in having some evidence for ambulatory BP measurements. I would like to draw your attention to some minor and major changes for the present format which you may accept if you feel necessary and appropriate. CLIP trials could give an input on the benefits of repeated BP measurements at community level and also the device used are more validated in PIH.
1. Introduction:
1a. Page 2, Line 61. It should be establish instead of established.
The sentence was changed due to Reviewer 2 suggestions.
1b. Are the abbreviations used standard abbreviations?
We added explanation for all non-standard abbreviations.
2. Materials and Methods:
2a. Duration of the study not mentioned
First sentence of study population section was changed:” This study was conducted for a period of 4 years and included 79 pregnant women with pre-existing primary hypertension and a documented history of eclampsia or pre-eclampsia in previous pregnancy/pregnancies”
2b. Line 69, Inclusion criteria were as follows: age over 18 years to be changed to Inclusion criteria were age over 18 years...
The sentence was corrected.
2c. Inclusion/ Exclusion Criteria: Looks incomplete, since the results does not include any nulliparous or women with first pregnancy. It looks like there is some study selection criteria which excludes these women. From Table 1- obstetric history, it is evident that women with 2nd, 3rd, 4th, 5th, 6th or 8th pregnancy were included in the study. In the introduction there is a mention of a Swedish study stating that risk of preeclampsia is more in first pregnancy. Kindly explain why women with first pregnancy were not included.
The aim of our study was to compare the results of BP measurements using various methods in pregnant women who had history of hypertension and eclampsia or pre-eclampsia during previous pregnancies. Of course it would be very interesting to examine women with current eclampsia, however, due to early prophylaxis and proper pharmacological treatment, the number of such females is limited.
2d. Even though BP measurements were taken every 5th week, why only three readings (10th, 25th and 37th week) were considered for analysis? Rationale behind this?
We decided to use the results of the blood pressure measurements at the end of each trimester for the greater clarity of the results. Showing the results of more blood pressure measurements would not contribute more to our study. A large number of BP measurements were made for other investigations.
2e. Are these devices from Omron and Spacelabs been validated and approved for measuring pregnancy induced hypertension?
Both devices were validated in pregnant women:
-
Van Den Heuvel JFM, Lely AT, Franx A, Bekker MN. Validation of the iHealth Track and Omron HEM-9210T automated blood pressure devices for use in pregnancy. Pregnancy Hypertens. 2019;15:37-41. doi:10.1016/j.preghy.2018.10.008
-
Shennan AH, Kissane J, de Swiet M. Validation of the SpaceLabs 90207 ambulatory blood pressure monitor for use in pregnancy. Br J Obstet Gynaecol. 1993;100(10):904-908. doi:10.1111/j.1471-0528.1993.tb15104.x
2f. For HBPM; what measures were taken to ensure that the BP measurement procedures were followed correctly? The two measurements in the morning and evening- were they done with some time interval between them? If so, how much?
Measurements were performed as recommended by ESC guidelines. Therefore in sentence: “Within 7 days before each medical visit, HBPM was performed with an oscillometric device” a statement was added: according to the ESC guidelines [6].
2g. Treatment: Was this a part of the study procedure? If so, what is its relevance in this study keeping in account the study objectives?
The use of antihypertensive drugs in the investigated females was necessary due to complications during previous pregnancies. In our opinion, the influence of drugs on the results of the study is not completely negligible. Therefore in limitations of our study we stated: “Additionally it was found that some factors may affect HBPM/ABPM differences, including age, BMI, history of cardiovascular disease, office hypertension, and antihypertensive treatment [21]. In particular, antihypertensive drug treatment may have independently increased the odds of diagnostic disagreement in our study.”
3. Results
3a. Pre-pregnancy data on weight and BMI. Was this self reported data by the women or was it measured? If measured, at what point were the women approached and included in the study?
Data on weight and BMI were obtained from obstetric records established by a gynecologists when it was found out that the female was pregnant.
3b. No mention of why women with first pregnancy not included in the study.
Please see section 2c.
3c. Figures: All these figures may not be required as the data is already presented in the Table 3. Please recheck if you can put only the relevant figures.
We agree that Bland-Altman analysis requires multiple tables and plots partially repeating the data which may be considered as redundant. Nevertheless, according to currently recommended standards of reporting Bland-Altman we are not able to resign from attaching both tables and figures (Abu-Arafeh, A., Jordan, H., & Drummond, G. (2016). Reporting of method comparison studies: a review of advice, an assessment of current practice, and specific suggestions for future reports. British journal of anaesthesia, 117(5), 569–575. https://doi.org/10.1093/bja/aew320)
According to the Reviewer No 1 suggestion we added additional information regarding Bland-Altman analyses and plots in captions of the figures and methodology section.
4. Discussion: Suggestion to improve the presentation in discussion section so as to appeal the readers and focusing more on the objectives of the study.
Both in the introduction and in the first section of the discussion, we justified the reasons for undertaking the research. We are afraid that a wider description of the methods of blood pressure measurement and their importance in diagnostics and treatment, especially in the pregnant women , would significantly prolong the already extensive discussion.